# Characterization and comparison of a 2-, 4- and 8-MHz central venous catheter ultrasound probe for venous air emboli detection

## Abstract

This paper presents a concept for detection of venous air emboli inside the superior vena cava using a central venous catheter with integrated Doppler ultrasound transducer installed on the tip. Several Doppler probes each with a single insonation frequencies of 2 MHz, 4 MHz or 8 MHz are characterized and compared for usefulness in this scenario. During in vitro experiments using an artificial blood circulatory with blood mimicking fluid bubbles with defined volumes were injected and recorded as gaseous embolic events. The in vitro results of measured embolus-blood-ratio values (EBR) in respect to the air bubbles volumes and its echogenicity showed a good correlation with the simulation model of spherical cross section scattering of such air bubbles. It is shown that the probe design still needs some improvements using a 4 MHz insonation frequency to get a useable detection sensitivity in such scenario within vena cava superior. The results suggest that it is possible to estimate the air bubble volume corresponding to the EBR using such a catheter probe.

**Keywords:** gaseous emboli, microemboli, central venous catheter, doppler ultrasound, emboli detection

Philipp Ch. Stark[1]
Christoph Kalkbrenner[1]
Werner Klingler[2,3]
Rainer Brucher[1]

1 Department of Mechatronics and Medical Engineering, University of Applied Science, Ulm, Germany

2 Anaesthesiology, SRH Kliniken,Sigmaringen, Germany

3 Ulm University, Experimental Anesthesiology, Ulm, Germany

## Zusammenfassung

In diesem Beitrag wird ein Konzept zur Erkennung von venösen Luftembolien in der oberen Hohlvene vorgestellt, bei dem ein zentraler Venenkatheter mit integriertem Doppler-Ultraschallwandler an der Spitze genutzt wird. Mehrere Doppler-Sonden mit jeweils einer einzelnen Beschallungsfrequenz von 2 MHz, 4 MHz oder 8 MHz werden charakterisiert und auf ihre Nützlichkeit in diesem Szenario verglichen. Bei in vitro Experimenten mit einem künstlichen Blutkreislauf mit Blutsimulatinsflüssigkeit wurden Blasen mit definierten Volumina injiziert und aufgezeichnet. Die in vitro Ergebnisse der gemessenen Embolus-Blood-Ratio-Werte (EBR) in Bezug auf die Luftblasenvolumina und ihre Echogenität zeigten eine gute Korrelation mit dem Simulationsmodell der Streuung solcher Luftblasen im Kugelquerschnitt. Es zeigt sich, dass das Design der Sonde bei Verwendung einer 4 MHz-Insonationsfrequenz noch einiger Verbesserungen bedarf, um eine brauchbare Detektionsempfindlichkeit in einem solchen Szenario innerhalb der Vena cava superior zu erhalten. Die Ergebnisse legen nahe, dass es möglich ist, die Luftblasenvolumina abzuschätzen, die der EBR mit einer solchen Kathetersonde entsprechen.

**Schlüsselwörter:** Gasförmige Embolien, Mikroembolie, Zentralvenöser Katheter, Doppler-Ultraschall, Embolie-Erkennung

## Introduction and objectives

Venous air embolism (VAE) is air entering the bloodstream via an opening at a venous blood vessel. If a venous blood vessel located above the heart is injured or opened during an operation, air will be sucked into the bloodstream due to negative hydrostatic pressure. Such a VAE is usually associated with neurosurgical operations in the (half) sitting position where VAE occurrence of up to 76% can be found [1], [2]. Due to case reports [3], [4] the theorized lethal air volume for adult humans is 200 to 300 ml, or 3–5 ml/kg and the entering rate of air can be assessed

up to 100 ml/sec. Therefore within 2–3 secs a lethal dose can be scored. A venous gaseous embolus (GE) can also enter the arterial blood stream via a patent foramen ovale (PFO) in the patient's heart. This for example can lead to a stroke or myocardial infarction. If air enters the cerebral circulation, only an amount of 2–3 ml is necessary to have a lethal effect and only 0.5–1.0 ml air can lead to a myocardial infarction from coronary air embolism [5].

For monitoring the heart for gaseous emboli during a surgery the reference standard is the transesophageal echocardiography (TEE).

But the TEE is the most sensitive monitoring systems for VAE since it is able to detect venous bolus doses as small as 0.02 ml/kg [6]. The TEE also has some downsides. Constant vigilance is required to detect GE. The TEE probe, which is inserted through the transesophageal rout, can cause mechanical injuries to the esophageal and gastric body. As well as thermal injuries due to the absorption of US energy by the tissue or vibrations inside the TEE probe [7].

Therefore, a central venous catheter (CVC) combined with an ultrasound Doppler probe was developed. The aim of this central venous catheter ultrasound (CVCUS) probe is to simplify the procedure of installing a monitoring probe and maintaining the insonation position and quality for recording and automated detecting air embolism during e.g., neurosurgical intervention. The CVCUS probe is designed to be located directly insight the superior vena cava (SVC) constantly monitoring passing gaseous emboli. With this CVCUS probe and its aspiration capability it would also be possible to suck out the air bubbles which often congregates in the atrium and exceeds a high-risk volume. To monitor and aspirate GE using only a TEE is not possible, a CVC would be needed. The CVCUS is capable of both features detecting and aspirating air volume. Single GE aspiration is difficult but GE accumulations such as air locks can be aspirated [8]. Prior to an operation using the CVCUS a PFO must be excluded, e.g., by means of Transthoracic Echocardiography (TTE). As shown in [9], [10] TTE in combination with contrast agent and a Valsalva maneuver is at least on par to TEE. TTE is non-invasive with a good diagnostic sensitivity. TTE can be used as an outpatient pre operation procedure.

The aim of this study was to assess the most useful insonation frequency for this specific application in simulations and experiments. On the one hand side the differences between the 2, 4 and 8 MHz CVCUS probe in their characteristics have to be confirmed and on the other hand side the ability, reliability and sensitivity to detect air emboli in blood mimicking fluid (BMF) [11] has to be shown in vitro studies and compared to theoretical simulations/calculations.

# Materials and methods

## Central venous catheter ultrasound probe

The presented ultrasound Doppler probe is a new design with a piezo crystal mounted on the distal tip of a CVC. The transducer was specially manufactured by MTB Medizintechnik Basler AG, Lindau, Switzerland. The piezo crystals diameter is dependent of the emitting frequency. Such a piezo crystal emits in a cone shaped geometry within a vessel (Figure 1).

Since the SVC has a mean length of approximately 70 mm and a mean diameter of 20 mm [12] the piezo crystal is topped with a dome shaped lens to achieve the appropriate expanded insonation diameter of 20 mm within a short distance. Through widening the insonation the whole SVC diameter can be monitored, and embolic events can be reliably detected. To fully cover the upper part of the Vena Cava the CVCUS probe must be inserted into the SVC in the standard procedure and the distal catheter tip positioned a couple of centimeters in front of the right atrium.

The beam characteristics of the CVCUS probes are examined using a motor-controlled hydrophone fixed at an xyz-scanner in a water bath. The CVCUS probes were scanned within a cross section perpendicular to the beam axis in an x-y-plane. In such a plane the distance from the hydrophone to the transducer was set to about 35 mm.

## Simulation

To find the insonation frequency best suited for detection of GE inside the SVC simulations/calculations of the Doppler sensitivity are made regarding the detectability of air bubbles within whole blood. Here for calculating the differential scattering cross section formula (1) in Figure 1 [13] was used to confirm the EBR in a given sample volume (SV).

For EBR calculations a SV with a cylindrical shape is used (Figure 2). The diameter of the assumed spherical GE is assumed with $D_e$=500 µm which corresponds to a volume of $V_e$=65 nl. This bubble size was chosen because GE with a diameter range from 67 µm to 900 µm are expected during SVC monitoring [14]. TEE imaging of the atrial area of single bubble events are showing similar sizing. The simulation of the EBR corresponding to embolus volume are not done in whole blood but in a similar medium like the BMF used in the in vitro experiments. The BMF was used at different concentrations and is stated with backscatter properties which are comparable to whole blood showing frequency spectrum properties with excellent correspondence [15]. The minimum detection limit was set to a 3 dB threshold, since a microemboli has to be at least 3 dB higher compared to the background signal as stated in [16].

$$\sigma_{E/B} = (\frac{c}{2\pi f})^2 \left| \sum_{m=0}^{\infty} (-1)^m (2m+1) a_m P_m(\cos\theta) i^{m+1} \right|^2 \qquad (1)$$

$$a_m = (-i)^m \frac{\rho_e Z_{b0} J_{m+1} J_{em} - \rho_b Z_{e0} J_m J_{em+1} + (\rho_b - \rho_e) m J_m J_{em}}{\rho_e Z_{b0} H_{m+1} J_{em} - \rho_b Z_{e0} H_m J_{em+1} + (\rho_b - \rho_e) m H_m J_{em}} \qquad (2)$$

$$Z_{b0} = \frac{2\pi f r_0}{c_b} \qquad (3)$$

$$Z_{e0} = \frac{2\pi f r_0}{c_e} \qquad (4)$$

$c_b, c_e$: *Ultrasound propagation velocity in blood and embolus*

$f$: *Insonated frequency*

$\rho_b, \rho_e$: *Mass density of the blood and embolus*

$P_m\{\cos\theta\}$: *Legendre polynomial of order m*

$\theta$: *reflection angle*

$r_0$: *Embolus radius*

$J_m\{Z\}$: *Spherical Bessel function of order m*

$H_m\{Z\}$: *Spherical Hankel function of order m*

$N$: *number of red blood cells in SV*

$$EBR = 10 log_{10} \frac{\sigma_E + N \cdot \sigma_B}{N \cdot \sigma_B}$$

**Figure 1: Formula**

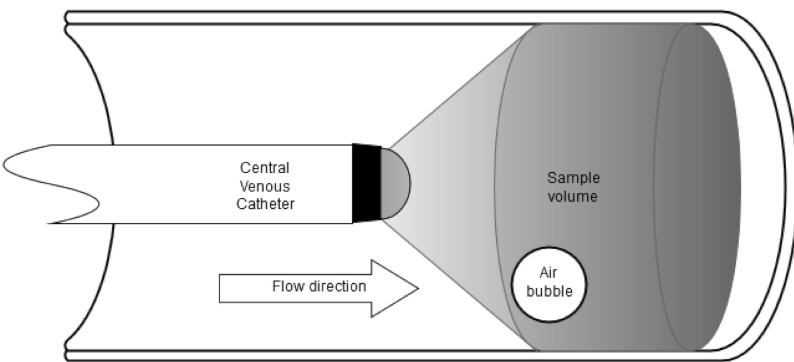

**Figure 2: Central venous Doppler catheter in blood vessel, its beam characteristic and sample volume positioning**

For all theoretical calculations and simulations, a BMF with a hematocrit value of 10% is used. In whole blood the 10% hematocrit values shows the same scattering coefficient as the 40% [17]. This value of 10% was therefore used for the experimental setups for better optical control based on acceptable transparency.

## In vitro studies

For the in vitro studies a modified version of the artificial blood circulatory (ABC) described in [18] was used. A tis-

sue model (MiniSim Trainer, Life-Tech, Inc., Houston, Texas, USA) followed by a self-designed optical control system was added after the modified SVC model (8 mm diameter for improved homogeneous insonation). A vertically installation of the main components for a smoother flow of the injected air bubbles was used.

A PC operated self-designed motorized bubble generator for well-defined gas injection was added to the ABC. This bubble generator consists of interchangeable calibrated high precision microliter syringes (Type: E115.1, E117.1, E119.1; Carl Roth GmbH + Co. KG, Karlsruhe, Baden-

Württemberg, Germany). The plunger is moved via a step motor (PD2-N4118L1804-2, Nanotec Electronic GmbH & Co. KG, Feldkirchen, Bavaria, Germany). The injection speed and volume are controlled via a PC with special implemented Software. The injected single bubbles are first detected by the CVCUS probe which is centrally situated inside the SVC model then detected by a sensitive transcranial Doppler 4 MHz probe located on the tissue phantom followed by an optical control system.

The CVCUS probes are axially placed inside the SVC model without any tilting towards the sides. The simulated hematocrit value of the BMF has about the same backscattering properties as whole blood with a hematocrit value of 42% to 48%. A BMF-hematocrit value of 10% is used for the experiments and simulations showing still transparency for optical control. Bubbles with three distinct volumes are injected into the ABC. The optical monitoring system (OMS) allows the fluid in the circulatory system to be monitored via an external high-definition camera (Basler pulse USB 3.0) with 50 fps through a window. To minimize data a region of interest (ROI) is used with the resolution of 960x960 px. For good image quality the window area was illuminated via an electroluminescent sheet which was placed beneath the window on the underside of the system.

For the reliable automatic detection of single GE events the enhancement in Doppler spectrograms (EBR measurements) of the 4 MHz tissue Doppler (Compumedics Germany GmbH, Singen, Germany), the catheter Doppler and the optical control system are used and compared. In respect to assess the catheters sensitivity an automatic detection algorithm is used in both Doppler spectrograms using wavelet based de-noising [19] and an adaptive threshold level for detection (Figure 3). The EBR is calculated using the wavelet de-noised I/Q Doppler data and a background (bgnd) signal which is calculated based on raw data extracted from the Dopplersystem. Due to denoising the blood signal is more or less removed which leads to a signal background level (bgng) around –10 dB. This bgnd audio signal, especially its amplitude, is based on the raw I/Q Doppler data which is filtered (Savitzky-Golay and Median filter) and occurring events are removed via clipping of the signal. In addition, the threshold for event trigger is calculated using the bgnd signal and a calculated EBR normalizing the raw data. This results in a threshold following the bgnd signal and showing a value around 0 dB referred to the intensity of the blood signal before denoising. This threshold can be shifted up and down for a more sensitive or less sensitive detection but keeping a margin for safe detection in mind.

Figure 4 shows the raw data audio signal on top. The corresponding calculated EBR and detection threshold below at about 0 dB. An event is detected when the EBR rises above the threshold which indicates an event enhancement. This example shows the start and end point of a detected event indicated by a triangle, respectively.

# Results

## Ultrasound probes and beam characteristics

Figure 5 shows a beam characteristic of a 2 MHz CVCUS probe. This CVCUS was scanned within a cross section perpendicular to the beam z-axis at a distance to the transducer of z=35 mm in x-y-plane on the left side. It shows a circular insonation area with a diameter of about 14 mm at the –6 dB limit. The right side shows x-, z-scan in the range from 15 cm to 31 cm. This scan shows a strong decreasing of intensity with increasing depth and therefore a strong opening of the beam due to the ultrasound lens on top of the piezo crystal.

The beam characteristics of a 4 MHz CVCUS probe are shown in Figure 6. The CVCUS was scanned at a distance to the transducer of 35 mm in x-y-axis. It shows a circular insonation area with a diameter of about 24 mm at the 6 dB limit. Furthermore, it shows a conical shape with an opening angle of 26° on the right side in the scan of the x, z-scan.

Figure 6 shows the beam characteristics of the 8 MHz CVCUS probe. This CVCUS was scanned at a distance to the transducer of 35 mm in x-y-axis, too. It shows a circular insonation area with a diameter of about 8 mm at the 6 dB limit (Figure 7 left). Here the x,z-scan shows a conical shape with an opening angle of 4.4° (Figure 7 right).

## Numerical simulation/calculation of cross section scattering and EBR

Due to the results of the CVCUS probe beam characteristics it is shown that the insonation cross section of two out of the three probes will not cover the diameter of the SVC. Therefore, a tube diameter of 8 mm is chosen for simulation and in vitro experiments so that the results are comparable within the same SV.

Now Figure 8 shows the absolute backscattered acoustical power of a single GE (black line) as a horizontal at low frequencies decreasing slowly at higher frequencies with resonance effects. The line of whole blood (red line) shows a logarithmic linear increasing with no resonances on increasing frequency, using a hematocrit value of 10%. All numerical calculations are based on formula (1) . For the calculation a SV with a cylindrical shape was used (sample volume length (SVL): 20 mm, sample volume diameter ($D_{sv}$): 8 mm). For sensitivity assessments the diameter of the GE was assumed with $D_e$=500 µm which has a spherical volume of $V_e$=65 nl. If the backscattered power of the blood is higher than the backscattered power of the GE, the event is detectable no longer. The vertical lines are marking the three different frequencies 2, 4 and 8 MHz.

At an insonation frequency of 8 MHz a GE with a diameter of $D_e$=500 µm ($V_e$=65 nl) and the mentioned SV is no longer detectable since the blood background signal (bgnd) succeeds the event intensity by 6.44 dB. At the

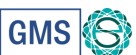

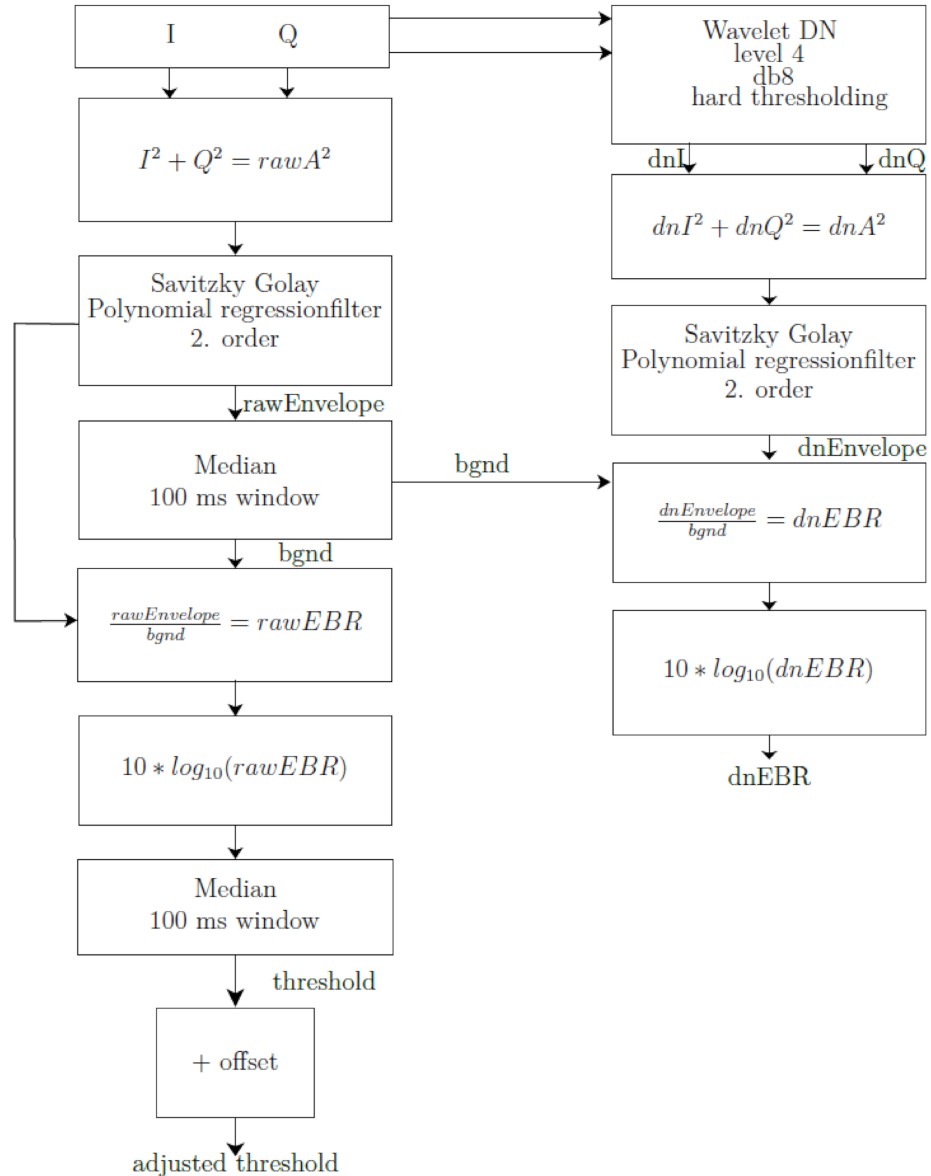

**Figure 3: Flow chart of the algorithm used for automatic GE detection in ultrasound Doppler spectrograms**

8 Mhz vertical line the ratio of the two square marked values leads to this EBR of an amount of –6.44dB. At 2 and 4 MHz the GE is still detectable, because the back-scattered power of the blood background signal is located below the GE intensity at –43.76 dB and –18.4 dB.

Figure 9 shows the EBR against size of GEs and therefore the theoretical detection limit of the 2 MHz, 4 MHz and 8 MHz Doppler. To determine the lower detection limit of the 2 MHz, 4 MHz and 8 MHz Doppler probes, a 3 dB threshold was used in respect to the Consensus Committee suggestion [16]. Since both the 2 and 8 MHz CVCUS probe have a beam characteristic cross section diameter smaller than the mean SVC diameter of 20 mm (see Figure 4 and Figure 6) a SV with the dimensions of $D_{sv}$=8 mm and axial SVL=20 mm is used for the simulations. The minimal detectable volume ($V_{xmin}$) is the crossing points of the horizontal 3 dB detection threshold line and the 2 MHz, 4 MHz and 8 MHz lines. With an insonation frequency of 2 MHz the $V_{2min}$=3.75 pl ($d_{2min}$=19

µm), with the 4 MHz insonation frequency $V_{4min}$=370 pl ($d_{4min}$=89 µm) and with the 8 MHz insonation frequency, $V_{8min}$=120 nl ($d_{8min}$=612 µm). The methodic limitation is shown at the resonance effect at the different insonation frequencies. At a frequency of 8 MHz such a resonance effect can be seen for GE larger than 1.2 µl ($d_{8max}$=1.32 mm). At an insonation frequency of 4 MHz the limitation is beginning at larger than 12 µl ($d_{4max}$=2.84 mm). The insonation frequency of 2 MHz shows its limitation at larger than 100 µl ($d_{2max}$=5.76 mm).

## In vitro studies for automatic detection of embolic events

The cross-section scans of the 2 and 8 MHz CVCUS probe showed that the detection area of the probes have a smaller diameter than the SVC, therefore a tube with the inner diameter of 8 mm was used for these experiments. An SVL of 20 mm in depth was used in all experiments.

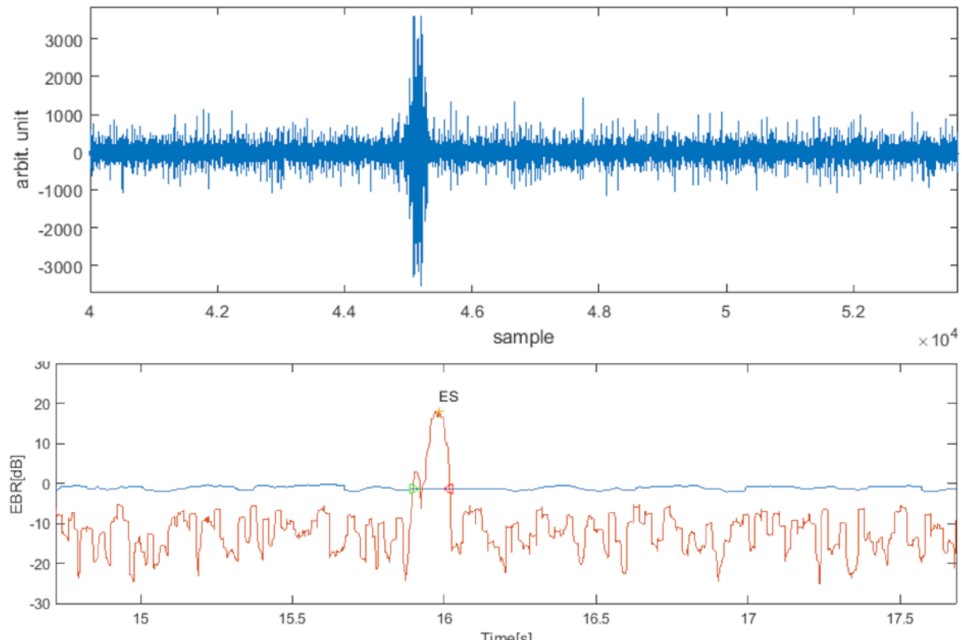

Upper: Background signal and GE event (ES) from the unprocessed raw Doppler audio signal. Lower: Wavelet de-noised EBR signal (orange) with adaptive threshold (blue line). Threshold is calculated using the background signal from the raw data audio signal. Green triangle: Start of GE event. Red triangle: End of GE event.

**Figure 4: Raw Doppler audio signal and denoised EBR signal**

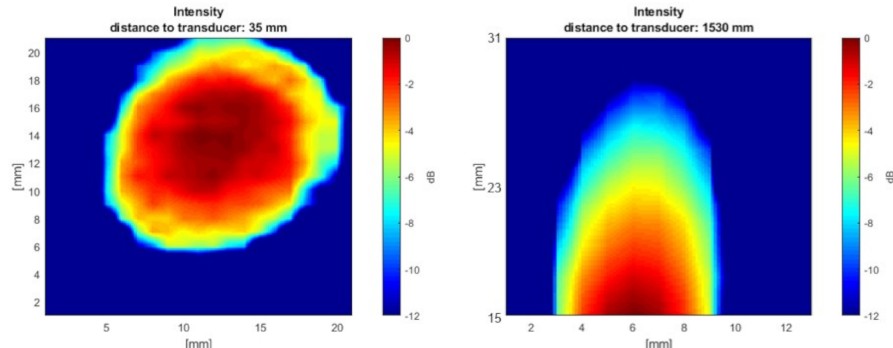

Left: Frontal xy-plane insonation cross section characteristic of 2 MHz probe at an axial distance of z=35 mm. Right: Transversal x,z-plane cross section insonation characteristic of the 2 MHz probe along the insonation beam z-axis with a range of distance to transducer from z=15 mm to z=31 mm.

**Figure 5: Beam characteristics of the 2 MHz CVCUS probe**

The injected bubbles have mean volumes of 37 nl (SD=±11 nl; $d_{37}$=413 µm), 105 nl (SD=±42 nl; $d_{105}$=585 µm) and 535 nl (SD=±127 nl; $d_{535}$=1 mm). For sensitivity analysis the detection threshold level could be set to 0 dB due to denoising, 3 dB lower than the suggested level in as an safety margin.

## Number of detections

Presented are the number of detected events in the Doppler signal recorded with the CVCUS probes compared to the number of bubbles counted by human observer using the optical control system as a reverence standard (Table 1). The automatic detection threshold was set to 0 dB. Due to the lower sensitivity the 8 MHz CVCUS probe the recorded data was additionally done with the

threshold set at –10 dB after denoising of background signal.

## EBR values simulation/calculation vs automatic detection

Values from Table 2 are combined with the simulation results in Figure 10. Here the EBR values of the detected GE are compared to the numerically calculated EBR value based on the equivalent volume at the different insonation frequencies 2, 4 and 8 MHz.

Comparing the Doppler data to the simulation results at 2 MHz insonation Figure 10 shows a deviation of –1.44 dB to +4.79 dB between measured EBR and numerical simulation (violet line in Figure 10). It also shows a non-linearity towards the smaller GE volumes.

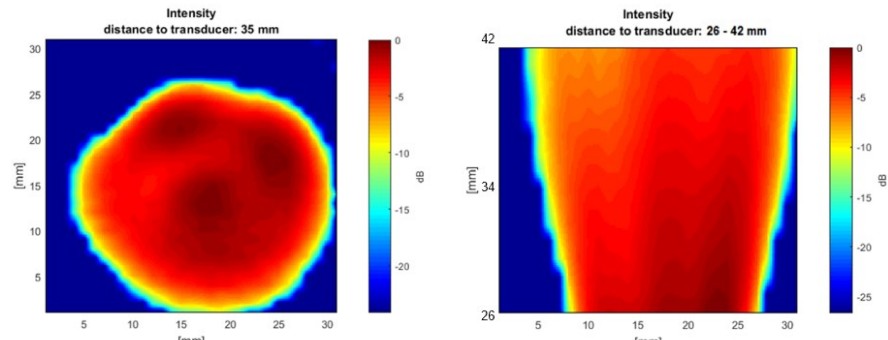

Left: Frontal x,y-plane insonation cross section characteristic of 4 MHz probe at a distance of z=35 mm. Right: Transversal x,z-plane cross section insonation characteristic of the 4 MHz probe along the insonation beam z-axis with a range of distance to transducer from z=26 mm to z=42 mm.

**Figure 6: Beam characteristics of the 4 MHz CVCUS probe**

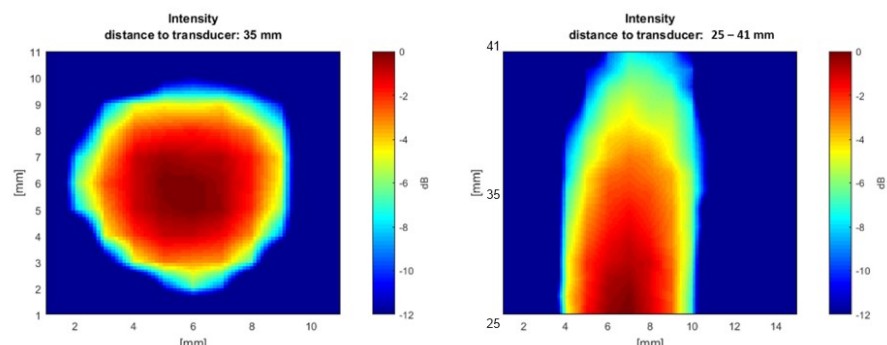

Left: Frontal x,y-plane insonation cross section characteristic of 8 MHz probe at a distance of z=35 mm. Right: Transversal x,z-plane cross section insonation characteristic of the 8 MHz probe along the insonation beam z-axis with a range of distance to transducer from z= 25 mm to z=41 mm.

**Figure 7: Beam characteristics of the 8 MHz CVCUS probe**

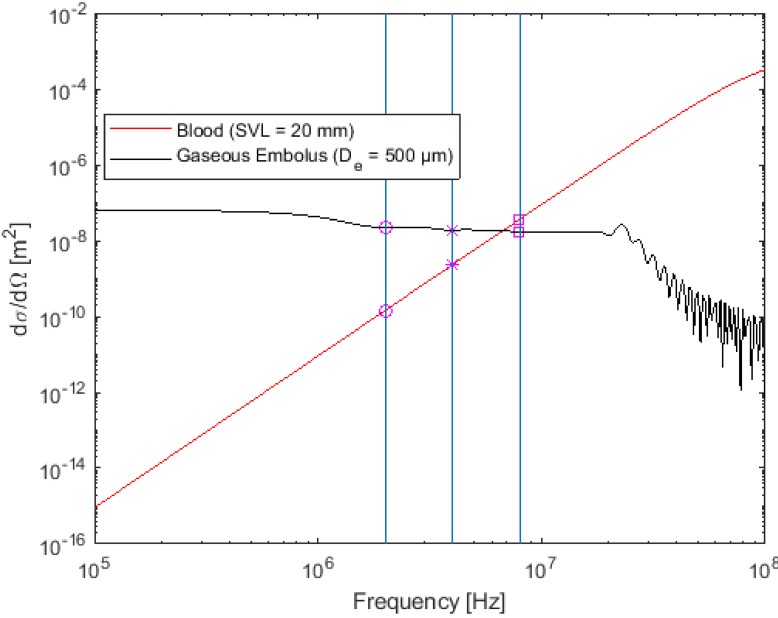

Black line: Gaseous embolus with a diameter of 500 µm (65 nl); Red line: Blood in the cylindrical sample volume (diameter d=8 mm); vertical blue lines: Indication for the used insonation frequencies of 2, 4 and 8 MHz.

**Figure 8: Numerical simulation/calculation of the absolute differential scattering cross section in whole blood in respect to EBR**

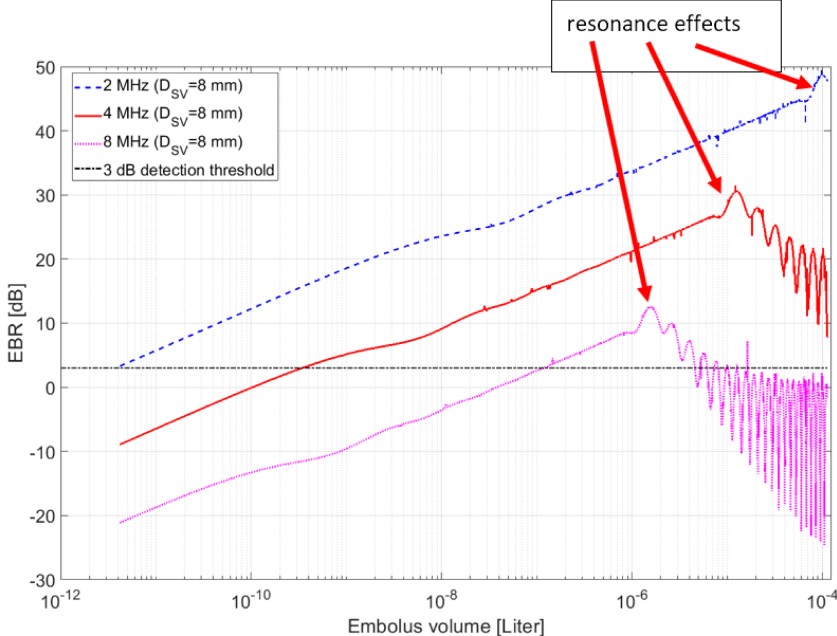

Dashed line: 2 MHz signal; solid line: 4 MHz signal; dotted line: 8 MHz signal; horizontal dash-dot line: 3 dB threshold according to the consensus committee [16] . EBR values above the threshold are detectable, below the threshold are undetectable. Minimal detectable embolus volume (Vxmin) at 2 MHz V2min = 3.75 pl, at 4 MHz V4min = 370 pl and at 8 MHz V8min = 120 nl

**Figure 9: Detection limit of the 2 MHz, 4 MHz and 8 MHz Doppler; simulation of EBR over the embolus volume in BMF**

**Table 1: Percentage of the automatic detected HITS via EBR threshold detection in relation to the manually counted number of bubbles by human observer as a reverence standard**

|  | 37 nl | 105 nl | 535 nl |
|---|---|---|---|
| **2 MHz** | 100% (38 of 38) | 96.55% (28 of 29) | 100% (22 of 22) |
| **4 MHz** | 100% (65 of 65) | 94.54% (52 of 55) | 84.21% (64 of 76) |
| **8 MHz (0 dB)** | 10% (5 of 50) | 25.64% (20 of 78) | 84.31% (43 of 51) |
| **8 MHz (−10 dB)** | 48% (24 of 50) | 65.38% (51 of 78) | 92.15% (47 of 51) |

**Table 2: Comparison of the simulated EBR and detected measured Mean-EBR values in dB for the injected bubble sizes at 2, 4 and 8 MHz**

| Volume | 37 nl | 37 nl SD=$\pm$11 nl | 105 nl | 105 nl SD=$\pm$42 nl | 535 nl | 535 nl SD=$\pm$127 nl |
|---|---|---|---|---|---|---|
|  | Simulation | Detection | Simulation | Detection | Simulation | Detection |
| **2 MHz** | 25.29 dB | 23.85 dB SD=$\pm$4.2 dB | 28.09 dB | 30.99 dB SD=$\pm$4.2 dB | 31.97 dB | 33.28 dB SD=$\pm$4.9 dB |
| **4 MHz** | 12.57 dB | 14.97 dB SD =$\pm$4.1 dB | 15.33 dB | 20.12 dB SD=$\pm$4.3 dB | 19.48 dB | 24.03 dB SD=$\pm$3.4 dB |
| **8 MHz (−10 dB)** | -0.04 dB | −2.46 dB SD=$\pm$4.8 dB | 2.7 dB | −0.57 dB SD=$\pm$5.2 dB | 7.19 dB | 8.59 dB SD=$\pm$5.0 dB |

The numerical EBR simulation and experimental EBR Doppler data at 4 MHz (green line in Figure 10) shows a difference of 2.4 dB to 4.95 dB and the similar non-linearity for the smaller bubble volumes as the 2 MHz CVCUS probe does.

Using the 8 MHz CVCUS probe EBR values are recorded with the fixed detection threshold level at −10 dB by applying denoising to the raw data of background signal due to the lower sensitivity. Here the experimental data showed a difference of 2.42 dB to 2.13 dB to the corresponding simulated numeral values (light blue line in Figure 10).

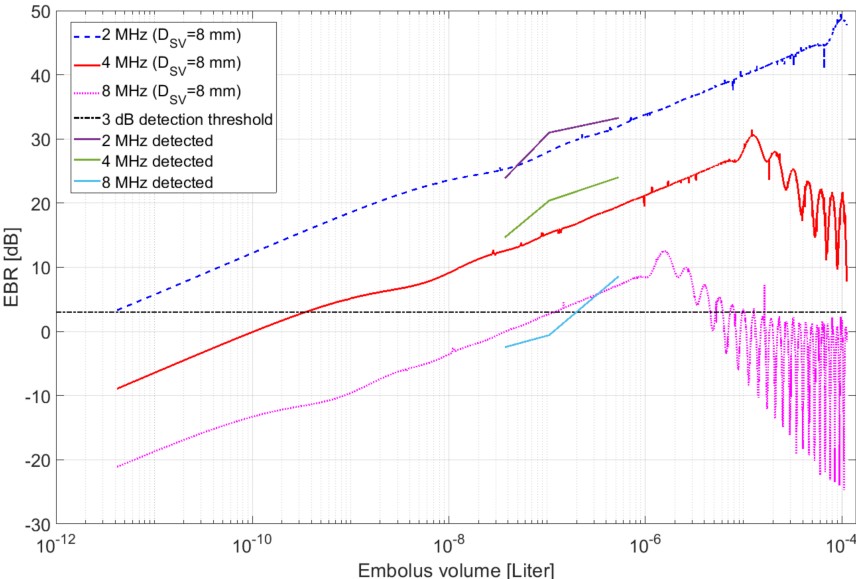

Comparison between experimental EBR data from Doppler (three diameters) and numerically simulated data (Fig.8). EBR Doppler data: violet (2 MHz), green (4 MHz) and light blue (8 MHz) solid short lines. Simulated EBR data: dashed blue (2 MHz), solid red (4 MHz) and dotted pink (8 MHz) lines.

**Figure 10: Comparison between experimental EBR data from Doppler (three diameters) and numerically simulated data (9)**

# Discussion

The principle aim of this study was to verify the functionality of the CVCUS concept, design and to find the best suited insonation frequency for an intravasal US monitoring via central venous catheter. The beam characteristics of the available/produced CVCUS probes were scanned in a water bath. The numerical simulation of the frequency dependent EBR – Embolus size shows the 8 MHz CVCUS probe as the least sensitive one and using 8 MHz only a very narrow range of embolus volumes can be detected compared to 4 MHz and 2 MHz. During in vitro experiments three types of air bubble diameter different in size were generated and injected into a circulatory system (Table 2). The injected bubbles were detected by using a CVCUS with insonation frequencies at 2, 4 and 8 MHz in addition with a 4 MHz tissue Doppler and an optical monitoring/control system.

Figure 6 shows the 4 MHz CVCUS probe fully covering the inner diameter of the SVC model in a 35 mm distance to the probe tip. Therefore, all GEs within the dynamic limits will be detected. A slight inhomogeneity in the cross section of the ultrasound beam can be observed which can result in a wider standard deviation of the EBR values and therefore also of the calculated embolus volumes.

The 2 and 8 MHz CVCUS have smaller opening angel in beam characteristics and a smaller insonation area (see Figure 4 and Figure 6) and therefore are not able to cover the whole inner area of the SVC with a quite homogenous insonation. Therefore, GEs near the wall can be missed respectively are not detected or measured with a lower echogenicity. In order to compensate for this effect and to detect all or most of the injected GEs in the experiments using all insonation frequencies, a smaller tube

diameter of 8 mm was chosen to cover in total the intravasal area. This change to a smaller vessel diameter was also considered in the numerical simulation shown in Figure 8 in discrepancy to the non-covering the normal diameter of the Vena Cava Superior.

The simulations and their corresponding experiments (Figure 8, Figure 9, Figure 10) show that the 2 MHz insonation frequency for the CVCUS Doppler probe is the most sensitive of the three applied frequencies. 2 MHz demonstrates in sensitivity a bottom limit of 3.75 pl and a top limit of 100 µl before entering the resonance effect showing ambiguity and therefore the limitation for size estimation by US echogenicity. Using the 4 MHz-CVUS probe the better total coverage for homogenous insonation of the vessel is given compared to the 2 MHz transducer. But this higher insonation frequency shows a lower point of entering the resonance effects and therefore a limitation for estimation of the embolus volume by EBR-measurements at about 10 µl. Here the sensitivity for volume detection is located at about 0.3 nl still a very high sensitivity compared to TEE at about 0.3 µl.

Using 8 MHz CVCUS probe suggests only a very limited detection range which is confirmed by the results stated in Table 1 and Table 2. This detection range can only be broadened by lowering the detection threshold level by de-noising. On the one hand side 10% of the 37 nl bubbles could be detected using a 0 dB detection threshold level. On the other hand, when the detection threshold level was lowered to – 10 dB, the detection rate was increased to 48% with 37 nl bubbles as sensitivity level (Table 1). These results also confirm the simulation of the 8 MHz EBR-Volume relation (Figure 10), where the 37 nl bubble is located at an EBR of – 0.04 dB (Table 2). But the upper limitation in size estimation is located at

about 1 µl and therefore at a range, which can be succeeded by air bubbles generated during neurosurgical intervention often.

Summarizing Figure 9 and Figure 10 shows an acceptable application range in size monitoring of bubbles for an insonation frequency at 4 MHz in respect to resonance effects and therefore to the methods limitation in size estimation and in regarding the sensitivity smaller than 1 µl. Although the detection range of 8 MHz insonation frequency can be expanded to smaller GE volumes e.g., higher sensitivity. The upper detection limit of 1.2 µl (d=1.3 mm) could be a problem since it is very close to the stated value of the 382 nl (d=0.9 mm). Since these values come from the decompression and not from surgical intervention, we can only take these sizes as a rough approximation.

The 2 MHz CVCUS probe has the widest detection range of the three probes. Looking at the lowest detection limit (at 3 dB according to [16]) of 3.75 pl. In an in-vivo scenario this might pose as a problem since even the smallest bubbles are detected. The probe is too sensitive if taken in consideration that according to Dexter et al. [20] a bubble of 1 nl volume only takes about five minutes to be fully absorbed.

The 4 MHz CVCUS probe has a less sensitive lower detection limit and a lower upper detection limit compared to the 2 MHz CVCUS probe. But with this lower detection limit of 370 pl the probe is still sensitive enough to detect bubbles which will be absorbed by the body in a short time. The 4 MHz CVCUS probe has the most homogeneous cross section of all the probes and is the only probe which is able to sufficiently insonate the SVC cross section (Figure 6). The overall CVCUS probe design, especially for 4 MHz, must be improved and redesigned so that the SVC can be insonated more homogeneously up to 20 mm across the entire diameter of a SVC.

This discussion on all the data is based on the experimental vessel size of 8 mm to keep it in more or less homogeneous insonation. Now if this vessel diameter will be extended to the normal size of SVC with 20 mm and an expanded insonation angel can be applied, an extension of the sample volume with the factor 6.25 is given corresponding to an increase of the background signal by about 8 dB. Then therefore, especially regarding the 4 MHz insonation frequency, the detection for air bubble will be positioned in the range between 10 nl up to 10 µl corresponding with air embolus diameter from 270 µm up to 2.7 mm. So, all interesting air bubbles can be detected in the SVC using a 4 MHz catheter probe.

The key advantage of the CVCUS over the TEE is twofold. Since the TEE cannot aspirate bubbles, a separate CVC must be placed. The CVCUS probe can perform both detection as well as aspiration. A PFO must be excluded preoperatively e.g., using a Transthoracic Echocardiography. The costs of a CVCUS probes are about 1.200 € and one probe can be reused for about 20 to 30 times. The TEE probes are about 10.000 € and can also be reused. Both probes must be sterilized after usage.

## Conclusions

The presented results suggest that it is possible to estimate the volume of detected single events of GEs with the CVCUS probe with insonating frequency of 4 MHz situated inside the bloodstream. Then detecting single gas bubbles could trigger an alarm if a certain summed up air volume level is reached counting all bubbles. To confirm this hypothesis additional in vitro experiments must be done regarding the variation in the detected EBR to a given bubble size in a correct sized vena cava model using also an improved CVCUS probe with widened ultrasound beam for more homogeneous insonation.

## Notes

Reviewer comments see Attachment 1.

## Acknowledgements

Research funding: This study is part of the project entitled "EmboKath" in cooperation with Compumedics Germany GmbH supported by the Arbeitsgemeinschaft industrieller Forschungsvereinigungen AiF (KF2186206KJ4). The authors would like to thank Compumedics Germany GmbH for their hardware, assistance and support.

## Conflict of interest

The authors declare that they have no competing interests.

## Attachments

1. attachment_1_hta000135.pdf (182 KB)
   Reviewer comments

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
