## [Reviewer comments · GMS Health Innovation and Technologies]

Reviewer comments:

1.) Please summarize the main findings of the study.

This study tries to find in vitro the reliability of the use of US in comparison to TEE in the detection of Air bubbles to prevent venous air emboli 19.) Please highlight the limitations and strengths. the main strength is that it can be a reliable and accurate technique to be used in contrast with the current Standard of care, the main limitation is that it is not discussed the acceptance by professionals of the new technique and its implications. Thus, it would be good to add such a consultation furthermore, when an invasive technique is used. It is not discussed also the consequences in terms of costs and safety issues. it is a very preliminary study also the fluid that they use differ from the characteristics of blood and that can affect the conclusions of future studies,...

2.) Please comment on the methods, results and data interpretation. If there are any objective errors, or if the conclusions are not supported, you should detail your concerns.

Regarding the methods and how they are discussed, the main concern would be the characteristics of the fluid used and how that can influence the results of the study. In reality blood is a fluid that is composed by cells and proteins that are embedded in a water based solution, so the density of the fluid and the characteristics of the different compounds should be beared in mind. On the other side there is no discussion on why the technique was chosen and which is the degree of acceptance by professionals.

3.) Please provide your detailed review report to the editor and authors.

The article is interesting in many senses although the degree of development of the technology is very low. It could not be ready for clinical practice in a short period of time. Apart from that there are some missing facts that should be considered on my view. On the one side, the abstract should referred in a more concrete way to the results and expressions such as good agreement should be avoided if data are not provided. on the other side, the introduction is sometime vague as there are some missing references specially regarding the frequency of the VAEs and also the differential costs and safety against the TEE. So expressions such as expensive and invasive monitoring should be avoided if they are not supported on evidence around the costs and when the alternative is also invasive (CVC is needed). It is also not discussed whether professionals could be in favour of the new technique in contrast to the current practice. It is very important professionals' acceptance and support in order to promote any technology that in theory is promising but in reality it can suppose issues from different points of view. Regarding the methods, one issue could be the diameter that has been used (this is discussed) and the type of fluid used. Obviously, this is in a phase of development considered as an early stage and thus no clinical comments would be needed but at least the implications and the comparisons against the current standard of care should be discussed. Furthermore, it would be good to consult with clinicians or even include them as authors because that could also improve the readability of the article for those not familiar with the technical aspects. It would be also good to have a discussion around the place in which the new technique could be used. In all cases, in cases in which TEE should be avoided, in cases in which VAE could be more frequent... In this regard there are some articles that could be useful to support the arguments: 1. Khan M, Schmidt DH, Bajwa T, Shalev Y. Coronary air embolism: incidence, severity, and suggested approaches to treatment. *Cathet Cardiovasc Diagn.* 1995;36:313–8. [PubMed] [Google Scholar]2. Dib

Attachment to: Stark PH, Kalkbrenner C, Klingler W, Brucher R. Characterization and comparison of a 2-, 4- and 8-MHz central venous catheter ultrasound probe for venous air emboli detection . *GMS Health Innov Techno.* 2022;16:Doc03. DOI: 10.3205/hta000135, URN: urn:nbn:de:0183-hta0001359. Available from: <https://www.egms.de/en/journals/hta/2022-16/hta000135.shtml>

J, Boyle AJ, Chan M, Resar JR. Coronary air embolism: a case report and review of the literature. Catheter Cardiovasc Interv. 2006;68:897–900. [PubMed] [Google Scholar]

4.) I recommend that the article.

needs some additional improvement according to my comments ready for publication should be rejected

After revision of the article, a recommendation for publication was made by the reviewers.

Attachment to: Stark PH, Kalkbrenner C, Klingler W, Brucher R. Characterization and comparison of a 2-, 4- and 8-MHz central venous catheter ultrasound probe for venous air emboli detection . GMS Health Innov Techno. 2022;16:Doc03. DOI: 10.3205/hta000135, URN: urn:nbn:de:0183-hta0001359. Available from: <https://www.egms.de/en/journals/hta/2022-16/hta000135.shtml>